# Interest on reserves, helicopter money and new monetary policy

**Duong Ngotran** *

Department of Business Administration, FPT University, Hanoi, Vietnam

* duongntt41@fe.edu.vn

## Abstract

We build a nonlinear dynamic model with currency, demand deposits and bank reserves. Monetary base is controlled by central bank, while money supply is determined by the interactions between central bank, commercial banks and public. In economic crises when banks cut loans, monetary policy following a Taylor rule is not efficient. Negative interest on reserves or forward guidance is effective, but deflation is still likely to be persistent. If central bank simultaneously targets both interest rate and money supply by a Taylor rule and a Friedman's k-percent rule, inflation and output are stabilized. An interest rate rule policy is just a subset of a more general monetary policy framework in which central bank can move interest rate and money supply in every direction.

**Data Availability Statement:** All code files are available on Figshare (DOI: 10.6084/m9.figshare.14256617.v1).

**Funding:** The author(s) received no specific funding for this work.

## Introduction

In the Great Recession and the recent COVID-19 crisis, the federal funds rate and the ECB policy rate have been near zero or even negative for a long time. Still, the deflation pressures have been so high in US and Europe. This phenomenon raised many concerns among economists and policy makers on how central bank's policy should be designed. Some possible solutions for raising inflation in this circumstance are forward guidance, helicopter money and quantitative easing [1, 2]. The last two tools can directly change money supply; however they cannot be a daily tool like an interest rate targeting policy in the standard monetary policy framework. This paper shows that central bank can add a Friedman's k-percent rule (money supply targeting) into the current standard framework (interest rate targeting) to hit the inflation target in economic crises.

The common traditional consensus among economists is that central bank cannot target both interest rate and money supply at the same time. Central bank chooses either monetary base as its main instrument [3–5] or the common interbank rate [6]. Nevertheless, the introduction of a new monetary policy tool—interest on reserves (IOR)—and the transformation of economy from making transactions with currency to electronic money will allow central bank to use both instruments. With IOR, the price of reserves might disconnect from the quantity of reserves in the banking system. With electronic money, money can earn a positive nominal interest rate. Therefore, it is a possible scenario that central bank can increase the level of reserves and raise rates at the same time.

**Competing interests:** The authors have declared that no competing interests exist.

This paper builds a dynamic model with bank reserves, currency, and demand deposits. The monetary base in our model is controlled by central bank, while the money supply is determined by the interactions between central bank, banks and public. The interbank rate is controlled either by open market operations or by adjusting IOR, so a wide range of conventional and unconventional monetary policies can be assessed in this model.

The main contribution of this paper is to characterize a new model of monetary policy when central bank can "simultaneously" and "actively" move money supply and interest rate in every direction. In textbooks, when money supply increases, interest rate goes down. However, in the electronic monetary system, if central bank wants, interest rate can go up when money supply increases. An interest rate rule policy (without manipulating money supply) is just a subset of a more general monetary policy framework.

By using this model, we still find that in normal times, an interest rate policy following a Taylor rule is an effective mean of controlling an economy. When central bank cuts rates, the amount of money supply increases because banks create more loans. As price is sticky in our model, outputs are stimulated in the short run due to the rise in the aggregate demand. The effect is identical to a standard New Keynesian model.

However, in economic crises when banks cannot make loans due to the capital constraints, a policy following a Taylor rule is insufficient for lowering real loan rates in our model. Even with a negative IOR or forward guidance, the outcome is only slightly better. The main reason is that the inflationary expectations also depend on the path of the money supply. Because of the decline in the endogenous money supply, our model shows that the deflation expectation happens in these economic crises. As a result of that, the real loan rate will be high even though the interbank rate touches the zero lower bound.

Targeting both money supply (by using helicopter money) and interest rate (by adjusting IOR) is very efficient in our model simulation. We find that central bank only needs to follow a simple Taylor rule and a Friedman's k-percent rule so that both outputs and inflation will be stabilized. After the crisis time, central bank can always come back to a simple Taylor rule alone.

## Related literature

Our model shares many similarities with a standard New Keynesian framework with the existence of a banking sector [7–11]. Banks play a role of intermediaries channeling funds from savers to borrowers in these models; while our model focuses on the function of creating money in the banking sector [12]. Banking crises create a liquidity problem for agents in the private sector as money supply declines. In this sense, our model is identical to [13], where a shock worsening the liquidity of pseudo-safe assets can create a crisis with a persistent deflation.

Our model also relates to a New Keynesian framework with money demand in the form of cash/deposit in advance [14–16]. Money (for households) could be either cash or deposits. Households need them in advance before making any transactions in the good and capital markets.

The salient feature separating our model from a standard New Keynesian is the money supply process and electronic payment system. Central bank in our model controls monetary base, but money supply is determined by the interactions between central bank, banks and public. [17, 18] characterize how central banks control reserves and manipulate the interbank rate, but the money supply part has not been explored in these papers.

To target both money supply and interest rate, central bank has to use IOR. This tool is already mentioned in the literature [19–24]. Our model, different from this line of research,

can connect IOR with banking reserves, money supply and interbank rate in a micro-founded dynamic setup.

We also discuss two unconventional monetary policies: negative IOR and forward guidance. The analysis of monetary policy with a negative interest rate can be found in [25]. Our model emphasizes the transmission through a negative IOR to the interbank rate and the deposit rate while the framework in [25] directly assumes that central bank can impose a negative short term rate. In both papers, the negative rate might be an important tool when the interbank rate is near zero. We also examine the effect of central bank's forward guidance policy [26–30] in a banking crisis context.

This paper extends the model in [31] to a more general environment, including both currency and electronic money. The model is very similar to a standard New Keynesian model, except the flows of electronic payments and the banker problem. Banking crises are the main themes in both papers and the effectiveness of different hypothetical polices are assessed under this context. However, [31] studies the inflation dynamics after the Great Recession and the appropriate policy to get out of the low interest rate environment. This paper focuses on a new type of monetary policy when central bank targets both money supply and interest rate by two common monetary policy rules.

## The environment

### Notation

Let $P_t$ be the price of the final good. We use lowercase letters to represent the real balance of a variable or its relative price. For example, the real reserves balance $n_t = N_t/P_t$, or the real price of intermediate goods $p_t^m = P_t^m/P_t$. The timing notation follows this rule: if a variable is determined or chosen at time $t$, it will have the subscript $t$. The gross inflation rate is $\pi_t = P_t/P_{t-1}$.

### Goods and production technology

We closely follow the model description in [31]. Our model extends [31] to an environment where currency and demand deposits coexist.

The model is in the deterministic setting and has six types of agents: bankers, households, wholesale firms, two types of retail firms (c-retailers and d-retailers), and a consolidated government.

There are four types of goods in the economy: cash-goods $y_{1,t}$ produced by c-retailers who only accept currency as the mean of payment, deposit-goods $y_{2,t}$ produced by d-retailers who only accept payment through banks, wholesale goods $y_t(j)$ produced by wholesale firm $j$ and intermediate good $y_t^m$ produced by households.

Each period households self-employ their labor $l_t$ to produce the homogeneous intermediate good $y_t^m$ under the production function:

$$y_t^m = l_t$$

Households sell $y_t^m$ to wholesale firms in the competitive market with the price $P_t^m$.

There is a continuum of wholesale firms indexed by $j \in [0, 1]$. Each wholesale firm purchases the homogeneous intermediate good $y_t^m$ from households and differentiates it into a distinctive wholesale goods $y_t(j)$ under the following technology:

$$y_t(j) = y_t^m$$

Wholesale firms faces the Rotemberg adjustment cost when they change their prices.

Two types of retail firms both produce the final good $y_{i,t}$ ($i = 1, 2$) by aggregating a variety of differentiated wholesale goods $y_t(j)$:

$$y_{i,t} = \left( \int_0^1 y_t(j)^{\frac{\epsilon-1}{\epsilon}} dj \right)^{\frac{\epsilon}{\epsilon-1}}, \quad i = 1, 2$$

As the markets for cash-goods and deposit-goods are perfectly competitive and they have the same constant return to scale production function, they have the same price $P_t$.

## Time, demographics and preferences

Time is discrete, indexed by $t$ and continues forever.

There is a measure one of identical infinitely lived bankers in the economy. Bankers discount the future with the discount factor $\beta$. Each period, they gain utility from consuming a basket $c_t$ that contains cash-goods $c_{1,t}$ and deposit-goods $c_{2,t}$. Their utility at the period $t$ can be written as:

$$\sum_{s=0}^{\infty} \beta^s \log(c_{t+s}), \quad \text{with } c_t = \left[ \sum_{i=1}^{2} \alpha_i^{\frac{1}{\sigma}} c_{i,t}^{\frac{\sigma-1}{\sigma}} \right]^{\frac{\sigma}{\sigma-1}}$$

where $\alpha_i$ is the share of cash-goods or deposit-goods in the basket and $\sigma$ is the elasticity of substitution between two goods in the basket.

There is also a measure one of identical infinitely lived households. Households discount the future with the discount factor $\tilde{\beta} < \beta$, so they will borrow from bankers in the steady state. This setup is similar to a model with two types of agents: patient and impatient households. Banks are owned by patient households, so impatient households borrow from banks. Banks will transfer profits to patient households.

Similar to bankers, each period households gain utility from consuming the basket $\tilde{c}_t$ and lose utility when providing labor $l_t$ to their own production. Household's utility at the period $t$ can be written as:

$$\sum_{s=0}^{\infty} \tilde{\beta}^s \left[ \log(\tilde{c}_{t+s}) - \chi l_{t+s} \right], \quad \text{with } \tilde{c}_t = \left[ \sum_{i=1}^{2} \alpha_i^{\frac{1}{\sigma}} \tilde{c}_{i,t}^{\frac{\sigma-1}{\sigma}} \right]^{\frac{\sigma}{\sigma-1}}$$

where $\chi$ is the weight of labor in the utility function.

Wholesale firms and retail firms are owned by households. The consolidated government includes both the government and the central bank, so for convenience, we assume there is no independence between the government and the central bank.

## Assets

There are two types of financial assets: bank loans to households $B_t^h$ and interbank loans $B_t^f$.

- **Bank loans to households** ($B_t^h$): We follow [14] to model the loan structure between bankers and households. The market for bank loan is perfectly competitive and the price of loan is $q_t^l$. When a household wants to borrow 1 dollar at time $t$, bankers will create an account for her and deposit $q_t^l$ dollars to her account. In the exchange for that, this household promises to pay $\delta_b, \delta_b(1 - \delta_b), \ldots, \delta_b(1 - \delta_b)^{n-1}, \delta_b(1 - \delta_b)^n \ldots$ dollars at time $t + 1, t + 2, \ldots t + n, t + n + 1 \ldots$ where $n$ runs to infinity and $0 < \delta_b \leq 1$ (Table 1). Loans are illiquid and bankers cannot sell loans.

**Table 1. Banker issues loans (left) and collects on loans (right).**

| Banker | | Banker | |
|---|---|---|---|
| Loans: $+S_t$ | Deposits: $+q_t^l S_t$ | Loans: $-\delta_b B_{t-1}^h$ | Deposits: $-\delta_b B_{t-1}^h$ |
| | Net worth: $+(1-q_t^l)S_t$ | | |

Let $B_t^h$ be the nominal balance of loan stock in the period $t$, let $S_t$ be the nominal flow of new loan issuance, we have:

$$B_t^h = (1 - \delta_b)B_{t-1}^h + S_t$$

- **Interbank loan** ($B_t^f$): Bankers can borrow reserves from other bankers in the interbank market. The nominal gross interest rate in the interbank market is the interbank rate $R_t^f$.

## Money and payment system

There are three types of money in our economy: currency $x_t$, zero-maturity deposits $m_t$ and reserves $n_t$.

- **Currency ($x_t$)**: is issued by central bank and held by households. If currency is held by bankers, it is automatically converted to reserves. Currency is used for transactions between households/bankers and c-retailers who sell cash-goods. Currency does not pay nominal interest. The amount of currency in circulation is endogenous in the equilibrium.

- **Zero maturity deposit (ZMD)** ($m_t$): is a type of e-money (IOU) issued by bankers. ZMDs have the same unit of account as currency. When holding these deposits, households earn a gross nominal interest rate $R_t^m$ which is determined by the perfectly competitive banking market. ZMDs are used for settling all transactions in the private sector, except for transactions between households/bankers and c-retailers. When the market between bankers and households open, household can convert ZMDs to currency or currency to ZMDs. They are insured by central bank, so they are totally safe.

- **Reserve** ($n_t$): is a type of e-money (IOU) issued by central bank for only bankers. It has the same unit of account with currency. Central bank pays a gross interest rate $R_t^n$ for these reserves. Interest on reserves $R_t^n$ is a monetary policy tool of central bank. At any moments, bankers can convert these reserves to currency and pay households. Reserves are used for settling transactions between bankers and bankers, bankers and the consolidated government.

Transactions with currency are simple. However, transactions with zero maturity deposits relates to many parties. We repeat the example in [31] to illustrate the connection between the flows of ZMDs and reserves. If wholesale firm A (whose account at bank A) pays 1 dollars to household B (whose account at bank B), the flows of payment will follow Table 2.

## Central bank and monetary policy

Central bank always uses the electronic payment system to conduct monetary policy. Each period, central bank transfers $\tau_t$ dollars to households' account. This can be seen as a shortcut of an open market operation process when central bank purchases government bonds from government. Then government transfers the payoffs to households. When $\tau_t$ is negative, it is equivalent to a lump-sum tax. For any transactions between central bank and households, as the payments are conducted through the banking system, we should think that they contain

**Table 2. Electronic payment system.**

| Wholesale firm A | | Household B | |
|---|---|---|---|
| Deposit: -1 | Payable:-1 | Deposit: +1 | |
| | | Receivable:-1 | |
| Bank A | | Bank B | |
| Reserve: -1 | Deposit: -1 | Reserve: +1 | Deposit: +1 |
| The Central Bank | | | |
| | Reserve (bank A): -1 | | |
| | Reserve (bank B): +1 | | |

**Table 3. Helicopter money / lump-sum tax.**

| The Central Bank | | Banks | | Public | |
|---|---|---|---|---|---|
| | Reserves: $+\tau_t$ | Reserves: $+\tau_t$ | Deposits: $+\tau_t$ | Deposits: $+\tau_t$ | Net worth: $+\tau_t$ |
| | Net worth: $-\tau_t$ | | | | |

two sub-transactions: one between central bank-bankers is settled by reserves, one between bankers—households is settled by ZMDs (Table 3).

## Timing within one period

1. Production takes place. Households sell goods to wholesalers, who, in turn, sell goods to retailers. All of the payments between households-wholesalers, wholesalers-retailers are delayed until the step 5

2. The cash-good market opens. Households need cash-in-advance to purchase from c-retailers. Bankers can convert reserves to cash to purchase from c-retailers.

3. The loan market between households and bankers opens. All the debt payments and loan issuance will be conducted electronically. The government transfers money to households. Households cannot exchange cash and deposits in this step.

4. The deposit-good market opens. Households need ZMD-in-advance to purchase goods from d-retailers. Bankers can create ZMDs to purchase d-goods.

5. Payments in the non-bank private sector are settled. Profits from firms are transfered back to households under either form of cash or ZMDs. Households can go to banks and readjust their portfolio between cash and deposits.

6. The banking market opens. Bankers can adjust the level of reserves by borrowing in the interbank market, receiving new deposits.

## Agents' problems

### Bankers

There is a measure one of identical bankers in the economy. These bankers have to follow central bank's regulations. There are three types of assets on a banker's balance sheet: reserves ($n_t$), loans to households ($b_t^h$), loans to other bankers ($b_t^f$). His liability side contains the zero-

maturity deposits that households deposit here ($m_t$).

| Banker | |
|---|---|
| Reserves: $n_t$ | Zero Maturity Deposits: $m_t$ |
| Loans to households: $b_t^h$ | Net worth |
| Loans to other bankers: $b_t^f$ | |

**Cost.** The banker faces a cost of managing loan, which is $\theta b_t^h$ in terms of deposit-goods.

The banker can adjust the level of his deposits and reserves after households and firms pay each other or when households withdraw currency from bank account. When these happen, the banker can witness that the deposits and reserves outflow from or inflow to his bank. Let $e_t$ be the net inflow of deposits and reserves going into his bank, he will treat $e_t$ as exogenous. When the banking market opens, as the deposit market is perfectly competitive, he can choose any amounts $d_t$ of deposit inflows or outflows to his bank.

In each period, the banker treats all the prices as exogenous and chooses $\{c_t, c_{1,t}, c_{2,t}, n_t, b_t^h, s_t, m_t, b_t^f, d_t\}$ to maximize his utility over a stream of consumption:

$$\max \sum_{t=0}^{\infty} \beta^t \log(c_t)$$

subject to

$$\frac{R_{t-1}^n n_{t-1}}{\pi_t} + \frac{R_{t-1}^f b_{t-1}^f}{\pi_t} + d_t + e_t + \tau_t = n_t + b_t^f + c_{1,t} \quad \text{(Reserve Flows)} \tag{1}$$

$$m_t = \frac{R_{t-1}^m m_{t-1}}{\pi_t} + q_t^l s_t + \theta b_t^h - \delta_b \frac{b_{t-1}^h}{\pi_t} + c_{2,t} + d_t + e_t + \tau_t \quad \text{(Deposit Flows)} \tag{2}$$

$$b_t^h = (1 - \delta_b) \frac{b_{t-1}^h}{\pi_t} + s_t \quad \text{(Loan Flows)} \tag{3}$$

$$c_t = \left[ \sum_{i=1}^{2} \alpha_i^{\frac{1}{\sigma}} c_{i,t}^{\frac{\sigma-1}{\sigma}} \right]^{\frac{\sigma}{\sigma-1}} \quad \text{(Consumption)} \tag{4}$$

$$n_t \geq \varphi m_t \quad \text{(Reserve Requirement)} \tag{5}$$

$$n_t + b_t^f + b_t^h - m_t \geq \kappa_t b_t^h \quad \text{(Capital Requirement)} \tag{6}$$

Eq (1) shows the change in reserves in the banker's balance sheet. After receiving the IOR, the previous balance of reserves becomes $R_{t-1}^n n_{t-1}/\pi_t$. He also collects payments from the inter-bank loans that he lends out to other bankers in the previous period $R_{t-1}^f b_{t-1}^f/\pi_t$. He can also increase his reserves by taking more deposits $d_t$. When doing that, his reserves and his liability increase by the same amount $d_t$ (Table 4). That is the reason we see $d_t$ appear on both the Eqs (1) and (2). The similar effect can be found on $\tau_t$- helicopter money and $e_t$. The banker treats $\tau_t$ and $e_t$ exogenously. Then, he can leave reserves $n_t$ at central bank's account to earn interest rate, or lend reserves to another bankers $b_t^f$. To purchase cash-goods $c_{1,t}$ from c-retailers, he converts reserves into currency (Table 5).During one period, his reserves balance can be temporarily negative. Still, in the end of every period, it must be positive and satisfies the regulation. Hence, the constraint in purchasing cash-goods implicitly lies in the reserve requirement.

**Table 4. Banker takes deposits (left) and makes interbank loan (right).**

| Banker | | Banker | |
|---|---|---|---|
| Reserves: $+d_t$ | Deposits: $+d_t$ | Reserves: $-b_t^f$ | |
| | | Interbank loan: $+b_t^f$ | |

**Table 5. Banker buys goods from c-retailers.**

| Central Bank | | Banker | | c-Retailers | |
|---|---|---|---|---|---|
| Reserves: $-c_{1,t}$ | Reserves: $-c_{1,t}$ | Net worth: | Currency: $+c_{1,t}$ | |
| Currency: $+c_{1,t}$ | (Vault Cash) | $-c_{1,t}$ | Inventory: $-c_{1,t}$ | |

Eq (2) shows the change in the banker's deposits. He makes loans to households by issuing deposits or creating ZMDs (Table 1). It is assumed that households have to pay loans from the account at the bank they borrow. The banker also issues his own ZMDs to purchase the consumption good from d-retailers ($c_{2,t}$) and to pay for the cost (in terms of deposit-goods) related to lending activities ($\theta b_t^h$) (Table 6). It is noted that he cannot create infinite amount of money for himself to buy consumption goods as there exists the capital requirement. Even without the capital requirement, because deposits are bankers' debts, the No-Ponzi condition is enough to prevent that action.

The banker faces two constraints in every period. At the end of each period, he has to hold enough reserves as a fraction of total deposits, which is showed in the Eq (5).

The second constraint is the capital requirement constraint. The left hand side of (6) is the banker's net worth (capital), which is equal to total assets minus total liabilities. We use the book value $B_t^h$ rather than the "market value" of loans $q_t^l B_t^h$ in the capital constraint. The reason is that illiquid bank loans should be treated differently from bonds. In reality, bank loans are often not revalued in the balance sheet when the interest rate changes. The constraint requires the banker to hold capital greater than a fraction of total loans in his balance sheet. We assume that $\kappa_t$ is a constant $\kappa$ in normal times. We later put the unexpected shock on this $\kappa_t$ to reflect the shock in a banking crisis.

Let $\gamma_t$, $\mu_t^r$ and $\mu_t^c$ be respectively the Lagrangian multipliers associated to the reserves flows, reserves constraint and the capital constraint. Let $r_t^h$ be defined as the real short-term lending rate. The first order conditions of the banker's problem can be written as:

$$\gamma_t = \left(\frac{\alpha_i c_t}{c_{i,t}}\right)^{1/\sigma} \frac{1}{c_t}, \quad i = 1, 2 \tag{7}$$

$$\gamma_t = \frac{\beta R_t^f \gamma_{t+1}}{\pi_{t+1}} + \mu_t^c \tag{8}$$

$$\gamma_t = \frac{\beta R_t^m \gamma_{t+1}}{\pi_{t+1}} + \mu_t^c + \varphi \mu_t^r \tag{9}$$

**Table 6. Banker buys goods from d-retailers.**

| Banker | | d-Retailers | |
|---|---|---|---|
| Deposits: $+(c_{2,t} + \theta b_t^h)$ | | Deposits: $+(c_{2,t} + \theta b_t^h)$ | |
| Net worth: $-(c_{2,t} + \theta b_t^h)$ | | Inventory: $-(c_{2,t} + \theta b_t^h)$ | |

$$\gamma_t = \frac{\beta R_t^n \gamma_{t+1}}{\pi_{t+1}} + \mu_t^c + \mu_t^r \tag{10}$$

$$(q_t^l + \theta)\gamma_t = \frac{\beta[\delta_b + (1 - \delta_b)q_{t+1}^l]\gamma_{t+1}}{\pi_{t+1}} + (1 - \kappa)\mu_t^c \tag{11}$$

$$r_t^h = \frac{\delta_b + (1 - \delta_b)q_{t+1}^l}{(q_t^l + \theta)\pi_{t+1}} \tag{12}$$

And two complimentary slackness conditions:

$$\mu_t^r \geq 0, \quad n_t - \varphi m_t \geq 0, \quad \mu_t^r(n_t - \varphi m_t) = 0 \tag{13}$$

$$\mu_t^c \geq 0, \quad n_t + b_t^f + (1 - \kappa_t)b_t^h - m_t \geq 0, \quad \mu_t^c(n_t + b_t^f + (1 - \kappa_t)b_t^h - m_t) = 0 \tag{14}$$

## Households

There is a measure one of identical households. These self-employed households produce homogeneous intermediate good $y_t^m$ to sell to the wholesale firms at price $P_t^m$. In each period, a household consumes cash-goods ($\tilde{c}_{1,t}$) from c-retailers and deposit-goods ($\tilde{c}_{2,t}$) from d-retailers.

Let $\tilde{B}_t^h$ be the nominal debt stock that she borrows from bankers. The loan structure follows the description in Table 1. There is an exogenous borrowing constraint for households with the debt limit $\tilde{b}_t^h \leq \overline{b^h}$.

In each period, households choose $\{\tilde{c}_t, l_t, a_t, x_t, \tilde{b}_t^h, \tilde{m}_t, \tilde{s}_t, \tilde{c}_{1,t}, \tilde{c}_{2,t}\}$ to maximize their expected utility:

$$\max \sum_{t=0}^{\infty} \tilde{\beta}^t(\log(\tilde{c}_t) - \chi l_t)$$

subject to

$$\text{CIA}: \quad \tilde{c}_{1,t} \leq \frac{x_{t-1}}{\pi_t} \tag{15}$$

$$\text{Loan Market}: \quad a_t + \delta_b \frac{\tilde{b}_{t-1}^h}{\pi_t} = \frac{R_{t-1}^m m_{t-1}}{\pi_t} + q_t^l s_t + \tau_t \tag{16}$$

$$\text{DIA}: \quad \tilde{c}_{2,t} \leq a_t \tag{17}$$

$$\text{Budget}: \quad \tilde{m}_t + x_t + \tilde{c}_{1,t} + \tilde{c}_{2,t} = a_t + \frac{x_{t-1}}{\pi_t} + p_t^m y_t^m + \frac{\Pi_t}{P_t} \tag{18}$$

$$\text{Production}: \quad y_t^m = l_t \tag{19}$$

$$\text{Loan flows}: \qquad \tilde{b}_t^h = (1 - \delta_b)\frac{\tilde{b}_{t-1}^h}{\pi_t} + \tilde{s}_t \tag{20}$$

$$\text{Constraint}: \qquad \tilde{b}_t^h \leq \overline{b^h} \tag{21}$$

$$\text{Consumption}: \qquad \tilde{c}_t = \left[ \sum_{i=1}^2 \alpha_i^{\frac{1}{\sigma}} \tilde{c}_{i,t}^{\frac{\sigma-1}{\sigma}} \right]^{\frac{\sigma}{\sigma-1}} \tag{22}$$

When the cash-good market opens, the household brings $(x_{t-1}/\pi_t)$ in cash to make transactions there. She faces the cash-in-advance constraint (15) when purchasing goods from c-retailers.

The loan market between bankers and households (Eq 16) only opens after that. Here the household pays a fraction of her old debts $(\delta_b b_{t-1}^h/\pi_t)$ and borrows new loan $(q_t^l s_t)$. All of the transactions are conducted electronically. We have assumed that she cannot readjust her portfolio between cash and deposits in this step. In the end, she brings $a_t$ amount of ZMDs to purchase goods from d-retailers.

Eq (18) is the household's general budget constraint. After receiving the profits $(\Pi_t/P_t)$ from wholesalers and revenue $(p_t^m y_t^m)$ from selling the intermediate good, she can go to banks and readjust her portfolio between deposits $(m_t)$ and currency $(x_t)$.

Let $\eta_{1,t}$, $\eta_{2,t}$, $\eta_t^b$ be the Lagrangian for the cash-in-advance, the deposit-in-advance and the borrowing constraint. Let $\lambda_t$ be the Lagrangian for the budget constraint.

$$\lambda_t + \eta_{i,t} = \left( \frac{\alpha_i \tilde{c}_t}{\tilde{c}_{i,t}} \right)^{1/\sigma} \frac{1}{\tilde{c}_t}, \qquad i = 1, 2 \tag{23}$$

$$p_t^m \lambda_t = \chi \tag{24}$$

$$\lambda_t = \frac{\tilde{\beta}(\lambda_{t+1} + \eta_{1,t+1})}{\pi_{t+1}} \tag{25}$$

$$\lambda_t = \frac{\tilde{\beta} R_t^m (\lambda_{t+1} + \eta_{2,t+1})}{\pi_{t+1}} \tag{26}$$

$$q_t^l(\lambda_t + \eta_{2,t}) = \frac{\tilde{\beta}[\delta_b + (1 - \delta_b)q_{t+1}^l](\lambda_{t+1} + \eta_{2,t+1})}{\pi_{t+1}} + \eta_t^b \tag{27}$$

And three complimentary slackness conditions:

$$\eta_{1,t} \geq 0, \qquad \frac{x_{t-1}}{\pi_t} - \tilde{c}_{1,t} \geq 0, \qquad \eta_{1,t}\left( \frac{x_{t-1}}{\pi_t} - \tilde{c}_{1,t} \right) = 0 \tag{28}$$

$$\eta_{2,t} \geq 0, \qquad a_t - \tilde{c}_{2,t} \geq 0, \qquad \eta_{2,t}(a_t - \tilde{c}_{2,t}) = 0 \tag{29}$$

$$\eta_t^b \geq 0, \qquad \overline{b^h} - b_t^h \geq 0, \qquad \eta_t^b(\overline{b^h} - b_t^h) = 0 \tag{30}$$

## Retail firms and wholesale firms

Following Rotemberg pricing, each wholesale firm $j$ faces a cost of adjusting prices, which is measured in terms of final good and given by:

$$\frac{\iota}{2} \left( \frac{P_t(j)}{P_{t-1}(j)} - \overline{\pi} \right)^2 y_t$$

where $\iota$ determines the degree of nominal price rigidity and $\overline{\pi}$ is the long-run inflation target. The wholesale firm $j$ discounts the profit in the future with rate $\tilde{\beta}^i \lambda_{t+i}/\lambda_t$. Her real marginal cost is $p_t^m$.

In a symmetric equilibrium, all firms will choose the same price and produce the same quantity $P_t(j) = P_t$ and $y_t(j) = y_t = y_t^m$. The optimal pricing rule then implies that:

$$1 - \iota(\pi_t - \overline{\pi})\pi_t + \iota\tilde{\beta} \left[ \frac{\lambda_{t+1}}{\lambda_t} \left( \pi_{t+1} - \overline{\pi} \right) \pi_{t+1} \frac{y_{t+1}}{y_t} \right] = (1 - p_t^m)\epsilon \qquad (31)$$

## The central bank and government

The consolidated government uses the payoffs from tax to pay for the IOR, then injects (withdraws) $\hat{\tau}_t$ amount of money to (from) households to target the interbank rate. All transactions are conducted through the banking system.

$$\tau_t = - \frac{(R_{t-1}^n - 1)n_{t-1}}{\pi_t} + \hat{\tau}_t \qquad (32)$$

In the conventional monetary policy, we assume that the IOR $R_t^n$ is fixed at a constant level $\overline{R^n}$. The interbank rate follows a common Taylor rule. To connect with the common New Keynesian literature, we assume that central bank does not want to have excess reserves in the banking system so they never set $R_t^f$ lower than $\overline{R^n} + \delta_f$ where $\delta_f > 0$. When the reserve requirement is no longer binding, there are infinite levels of reserves that can satisfy the interbank rate at its lower bound. In this case, we need a rule governing the motion of reserves and change the standard Taylor Rule. Later, we will relax the assumption and examine the situation when the banking system is awash of excess reserves and central bank controls the interbank rate by adjusting $R_t^n$.

The conventional monetary policy rule can be described as:

$$R_t^n = \overline{R^n} \qquad (33)$$

$$R_t^f = \max \left\{ \frac{\overline{\pi}}{\beta} \left( \frac{\pi_{t+1}}{\overline{\pi}} \right)^{\phi_\pi}, \quad \overline{R^n} + \delta_f \right\} \qquad (34)$$

## Equilibrium

***Definition***: *A perfect foresight equilibrium is a sequence of bankers' decision choice $\{c_t, c_{i,t}, n_t, b_t^h,$ $s_t, m_t, b_t^f, d_t\}$, household's choice $\{\tilde{c}_t, \tilde{c}_{i,t}, \tilde{b}_t^h, \tilde{s}_t, \tilde{m}_t, x_t, l_t, y_t^m\}$, the firms' choice $\{y_t\}$, central bank' choice $\{\tau_t, R_t^n\}$, and the market price $\{q_t^l, R_t^f, \pi_t, p_t^m\}$ such that:*

1. *Given the market price, the initial conditions and central bank's choices, banker's choices solve the banker's problem, household's choices solve the household's problem, firm's choice solves the firm's problem.*

2. *All markets clear*:

$$Net\ inflows\ of\ deposits: \quad d_t + e_t = -\left(x_t - \frac{x_{t-1}}{\pi_t} - c_{1,t}\right) \tag{35}$$

$$The\ interbank\ market: \quad b_t^f = 0 \tag{36}$$

$$Total\ ZMDs: m_t = \tilde{m}_t$$

$$Loan\ Market: b_t^h = \tilde{b}_t^h$$

$$Good\ Market: \quad y_t = \sum_{i=1}^{2}(c_{i,t} + \tilde{c}_{i,t}) + \theta b_t^h + \frac{\iota}{2}(\pi_t - \overline{\pi})^2 y_t \tag{37}$$

Later, we set some different central bank's monetary polices subject to a set of equations in this perfect foresight equilibrium. In each case, we might also change the set of central bank's monetary policy tools. For convenience, we define **(AD)** as the set of equations in the perfect foresight equilibrium, excluding the monetary policy and exogenous shocks.

**Definition**: *Let **(AD)** contain the set of equations and conditions in* (44)–(66) *(Mathematical Appendix)*.

## Theoretical results

We make the following assumption to ensure that households borrow from bankers in the steady state.

**Assumption 1**. *The discount factors of bankers and households satisfy*:

$$\frac{\beta\delta_b - \theta\overline{\pi}}{\overline{\pi} - \beta(1 - \delta_b)} > \frac{\tilde{\beta}\delta_b}{\overline{\pi} - \tilde{\beta}(1 - \delta_b)}$$

The next assumption ensures that inflation is equal to central bank's inflation target in the steady state.

**Assumption 2**. *The monetary policy tools satisfy*:

$$\lim_{t\to\infty}\frac{\hat{\tau}_t}{n_t + x_t} = \overline{\pi} - 1\overline{\pi}$$

$$\overline{R^n} + \delta_f < \frac{\overline{\pi}}{\beta}$$

We start with the first result showing the relationship between the interest on reserves, the deposit rate and the interbank rate.

**Theorem 1**. *In equilibrium*:

- *The lower bound of the interbank rate and the deposit rate is the interest on reserves. In all cases,* $R_t^n \leq R_t^m \leq R_t^f$

- *When the constraint of reserve requirement is not binding,* $R_t^f = R_t^m = R_t^n$.

The benefits of holding reserves come from two sources. First, bankers earn the interest on reserves. Second, bankers satisfy the reserve requirement, showing in the shadow price of reserve constraint $\mu_t^r \geq 0$. When the banking system has a huge amount of excess reserves,

second benefit is no longer there $\mu_t^r = 0$, and the interbank rate is equal to the interest on reserves.

**Theorem 2**. *In equilibrium, the monetary base, as sum of reserves and currency in circulation, is decided solely by central bank*:

$$\frac{n_{t-1} + x_{t-1}}{\pi_t} + \hat{\tau}_t = n_t + x_t \tag{38}$$

When households withdraw money from their bank accounts, it only changes the level of reserves but does not affect the level of monetary base. We assume that central bank will target the interbank rate, so it implies that central bank will never leave the banking system with the negative amount of reserves.

**Theorem 3**. *Under the Assumption (1)-(2) and if $\kappa$ satisfies*:

$$\kappa < 1 - \frac{(1 - \varphi)\overline{m}}{b^h}$$

*where $\overline{m}$ is the steady state value of m, then there exists a unique steady state. Moreover, in this steady state, the reserve requirement is binding while the capital constraint is not binding.*

This unique steady state reflects the banking systems in US and EU before the Great Recession. There were no excess reserves, and the policy rates were higher than 2 percent. These central banks' main tool was open market operations at that time, rather than the IOR. After the Great Recession, due to many rounds of quantitative easing (unconventional monetary policy), the excess reserves skyrocketed. However, in the long term, these central banks have a plan to scale down their balance sheets' size to the pre-crisis level. As a result of that, this unique steady state might properly show the long-term position of those central banks.

## Numerical experiments

### Calibration

Data are calibrated to match the US economy before the Great Recession (Table 7). Each period in the model is equivalent to one quarter. The bankers' discount factor is set to the standard value 0.99. The reserve requirement is calibrated to reflect the ratio between the total level of reserves and the total ZMDs. In Dec 2007, before the financial crisis, the total level of reserves was around 9 billion dollars. The total MZM (Money Zero Maturity) was approximately 8130 billion, 75-80 percent of which are checkable deposits, saving deposits and money market deposit accounts. The level of ZMDs was therefore 6000 billion, and the ratio between reserves and ZMD is 0.0015. We round up this ratio to 0.002 and set $\varphi$ = .002. The monitoring cost and the loan amortization factor are set exogenously. The risk weight is calibrated so that it satisfies the conditions in the Theorem 3 to ensure the unique steady state. For $\kappa \geq 0.2$, the capital constraint is binding in the steady state. Therefore, we set $\kappa$ at 0.18.

The consumption basket is calibrated to match the ratio between currency and ZMDs in the economy. In Dec 2007, the total level of currency was 760 billion. [32] estimates that more than half of US dollar bills are held overseas, so we end up with around 330 billion in currency held by US public. At the same time, ZMDs was 6000 billion, so currency accounts for approximately 6 percent of the total money supply (MZM). We calibrate $\alpha_1 = 0.06$ and $\alpha_2 = 0.94$ to match this fact. The elasticity of substitution between cash goods and deposit goods is set exogenously to 10.

For central bank's parameters, the only unusual parameter is the interest on reserves. We set it at 25 basis points and consider it as the lower bound for IOR at most cases in our

**Table 7. Parameter values.**

| Param. | Definition | Value |
|---|---|---|
| *Bankers* | | |
| $\beta$ | Banker's discount factor | 0.99 |
| $\varphi$ | The reserve requirement | 0.002 |
| $\kappa$ | The risk weight | 0.18 |
| $\theta$ | The monitoring cost | 0.0005 |
| $\delta_b$ | Loan amortization | 0.5 |
| *Households* | | |
| $\tilde{\beta}$ | Household's discount factor | 0.985 |
| $\chi$ | Relative Utility Weight of Labor | 0.586 |
| $\overline{b^h}$ | The borrowing limit | 1.5 |
| *Consumption Basket* | | |
| $\alpha_1$ | Share of cash goods in the basket | 0.06 |
| $\alpha_2$ | Share of deposit goods | 0.94 |
| $\sigma$ | Elasticity of substitution between two goods | 10 |
| *Firms* | | |
| $\epsilon$ | Elasticity of substitution of wholesale goods | 4 |
| $\iota$ | Cost of changing price | 80 |
| *Central bank* | | |
| $\overline{\pi}$ | Inflation long-run target | 1 |
| $\phi_\pi$ | Policy responds to inflation | 1.25 |
| $\overline{R^n}$ | The constant IOR | 1+0.25/400 |
| $\overline{R^n} + \delta_f$ | The lower bound for FFR | 1+0.251/400 |

quantitative exercises. All other parameters are in the standard range in the New Keynesian literature.

## Shock on the interbank rate

The first numerical experiment is to examine the response of the economy when central bank cuts the interbank rate. The list of equations for monetary policy and exogenous shock is:

$$R_t^f = \max\left\{ \frac{\overline{\pi}}{\beta}\left(\frac{\pi_{t+1}}{\overline{\pi}}\right)^{\phi_\pi} \exp\left(u_t^f\right), \quad \overline{R^n} + \delta_f \right\}$$

$$R_t^n = \overline{R^n} \tag{M1}$$

$$u_t^f = \rho_f u_{t-1}^f, \quad u_0^f \text{ is given}$$

From the steady state, there is an unexpected shock on the interbank rate $u_0^f$, then agents know the shock will die slowly with the persistence $\rho_f$. These two parameters' values are in Table 8. We can summarize this problem as (**P1**) containing the set of conditions in (**AD**) and (**M1**). Fig 1 shows the response of the economy under this experiment.

When central bank cuts the interbank rate by increasing the level of reserves, $q^l$ increases and the real lending rate $r^h$ is lower for households. As bankers lend out by creating money under the form of ZMDs, the money supply increases. It is noting that both currency and ZMDs go up after this shock. The aggregate demand is stimulated and inflation goes up.

**Table 8. Parmeter values in experiments.**

| Param. | Definition | Value |
|---|---|---|
| *Experiment* (M1) | | |
| $u_0^f$ | Initial interest shock | -2/400 |
| $\rho_f$ | The persistence of the interest shock | 0.7 |
| *Experiment* (M2) | | |
| $\kappa_0$ | Initial shock on the risk weight | 0.224 |
| $\rho_\kappa$ | The persistence of shock | 0.95 |
| *Experiment* (M3) | | |
| $\underline{R}^n$ | The negative lower bound for IOR | 1-0.35/400 |
| $T_e$ | Number of periods $\underline{R}_t^n = \underline{R}^n$ | 50 |
| *Experiment* (M4) | | |
| $\overline{\epsilon}$ | The forward guidance signal | -1/400 |
| $T_{FG}$ | Number of periods in forward guidance | 16 |
| *Experiment* (M5) | | |
| $\phi_m$ | Coefficient in mixed rule | 0.2 |
| *Experiment* (M6) | | |
| $\alpha_n$ | Coefficient in Taylor Rule | 0.8 |
| $T_e$ | Number of periods targeting both MS and IR | 50 |

Basically, this is identical to the reaction in a standard New Keynesian model. The only key difference here is the role of commercial banks in creating money. Money supply is totally endogenous and depends on the interaction between central bank, banks and public. Another crucial point should be noted that the effect of monetary policy, in this conventional setting, depends on the transmission from the interbank rate to the lending rate in the loan contract between bankers and households.

## Financial crisis—Taylor rule

We examine a simple form of a banking crisis by imposing an exogenous shock on $\kappa_t$. This reflects an increase in the bad loans that causes the capital constraint to bind. Central bank is assumed to respond to this crisis by using a Taylor Rule.

$$R_t^f = \max\left\{\frac{\overline{\pi}}{\beta}\left(\frac{\pi_{t+1}}{\overline{\pi}}\right)^{\phi_\pi}, \quad \overline{R^n} + \delta_f\right\}$$

$$R_t^n = \overline{R^n} \tag{M2}$$

$$\kappa_t = \rho_\kappa \kappa_{t-1} + (1 - \rho_\kappa)\overline{\kappa}, \quad \kappa_0 \text{ is given}$$

From the steady state, there is an unexpected shock $\kappa_0$. After that, the shock dies with the persistence $\rho_\kappa$. These two parameters' values are reported in Table 8. We can summarize this problem as (**P2**) containing the set of conditions in (**AD**) and (M2).

As the lower bound on the interbank rate is the IOR, the Taylor rule is constrained by the IOR. To see whether the negative IOR helps central bank, we conduct another similar

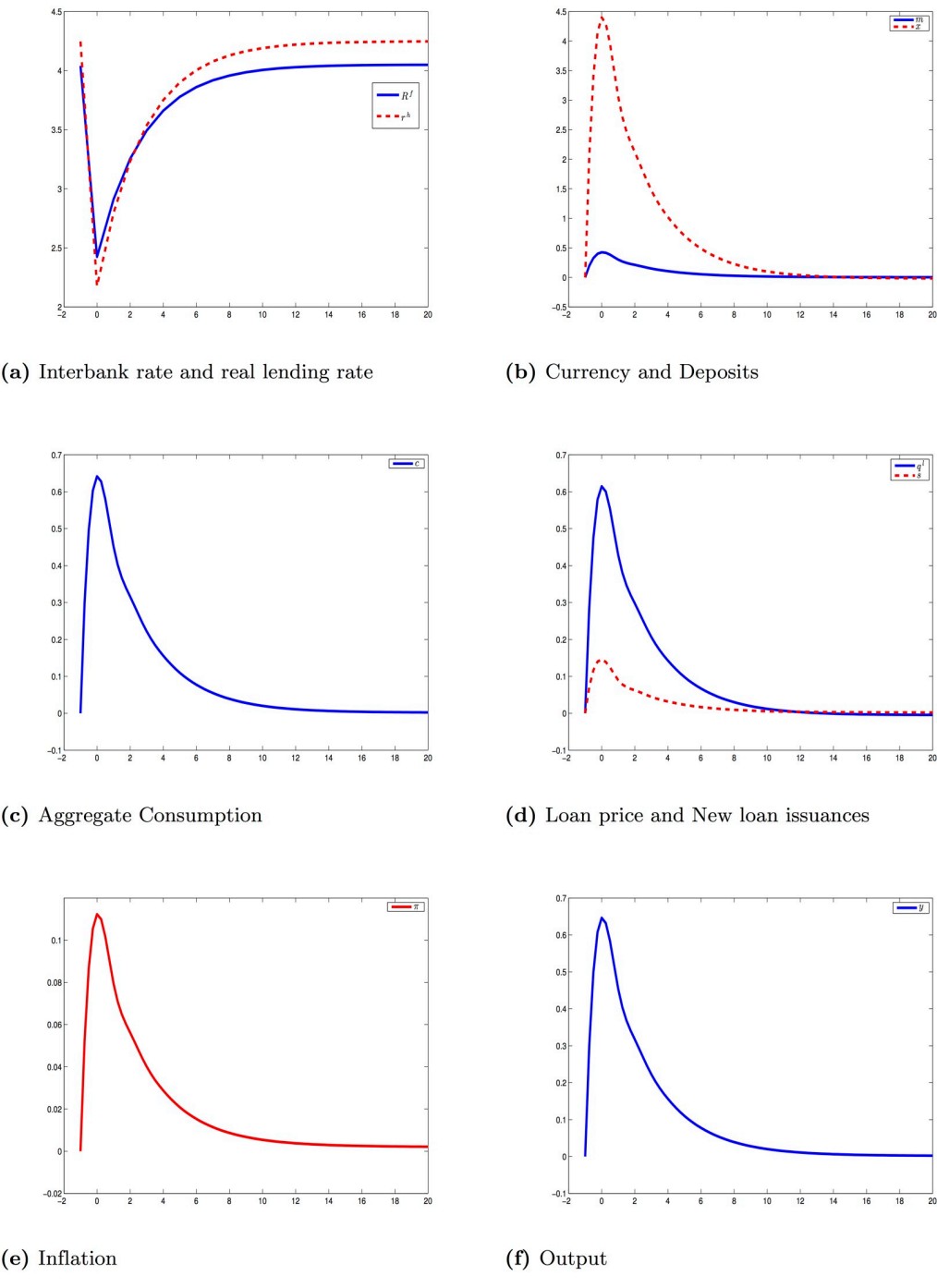

**(a)** Interbank rate and real lending rate

**(b)** Currency and Deposits

**(c)** Aggregate Consumption

**(d)** Loan price and New loan issuances

**(e)** Inflation

**(f)** Output

**Fig 1. Impulse response to interest rate shock (M1).**

experiment but allow the interest on reserves is negative during $T_e$ periods.

$$R_t^f = \max\left\{\frac{\overline{\pi}}{\beta}\left(\frac{\pi_{t+1}}{\overline{\pi}}\right)^{\phi_\pi}, \quad \underline{R^n} + \delta_f\right\}$$

$$R_t^n = \begin{cases} \underline{R^n}, & \text{if } t \leq T_e \\[2mm] \overline{R^n}, & \text{if } t > T_e \end{cases} \tag{M3}$$

$$\kappa_t = \rho_\kappa \kappa_{t-1} + (1 - \rho_\kappa)\overline{\kappa}, \quad \kappa_0 \text{ is given}$$

According to [2], the Fed estimates that the interest rate paid on bank reserves in the U.S. could not practically be brought lower than about -0.35 percent to avoid the bank withdrawal. Hence, we set $\underline{R^n} = 1 - 0.35/400$ and $T_e = 50$. We can set the problem (**P3**) contains the equations in (**AD**) and (M3). Fig 2 shows the response of the economy under these two experiments. Here are some important remarks:

- A Taylor rule with a negative IOR is more efficient than the one with zero lower bound. [25] gets the same result from a standard New Keynesian framework. However, the positive effect is very small at dealing with this type of financial crisis.

- The banking crisis is dangerous as central bank cannot rely on the pass-through from the interbank rate to the prime rate any more. In our simulation, the interbank rate is at its lower bound for 12 quarters with the IOR at 25 basis points and 4 quarters with the IOR at -35 basis points; however, the real lending rate still goes up. When the capital constraint is binding $\mu_t^c > 0$, the wedge between the interbank rate and the prime rate must reflect this shadow price.

- The banking crisis is often accompanied by the deflation episode and the insufficient demand. Bankers cut loans; therefore, the money supply plummets even though the monetary base increases. In our model, the level of currency goes up a little bit during the crisis. The deposit rate is near zero or even negative in our two experiments, making currency more favorable in households' eyes. However, this change does not affect much the total money supply because currency only accounts 6 percent of the money supply in the steady state.

- Lacking liquidity, households cut their own consumption and output declines. A Taylor rule, even with a negative IOR, is not enough for simulating the aggregate demand in this case. When the link between the interbank rate and the prime rate breaks, the conventional monetary policy is generally not effective.

## Forward guidance

The recent literature in monetary economics focuses on a forward guidance policy when an interest policy is restricted by the zero lower bound. In this section, we do a simple experiment to see whether a forward guidance policy is useful in the above crisis. There are two common ways to model how central bank informs public about the interest rate path in the future: (i) interest rate peg and (ii) news shock on a Taylor rule. We follow the latter in [30] to

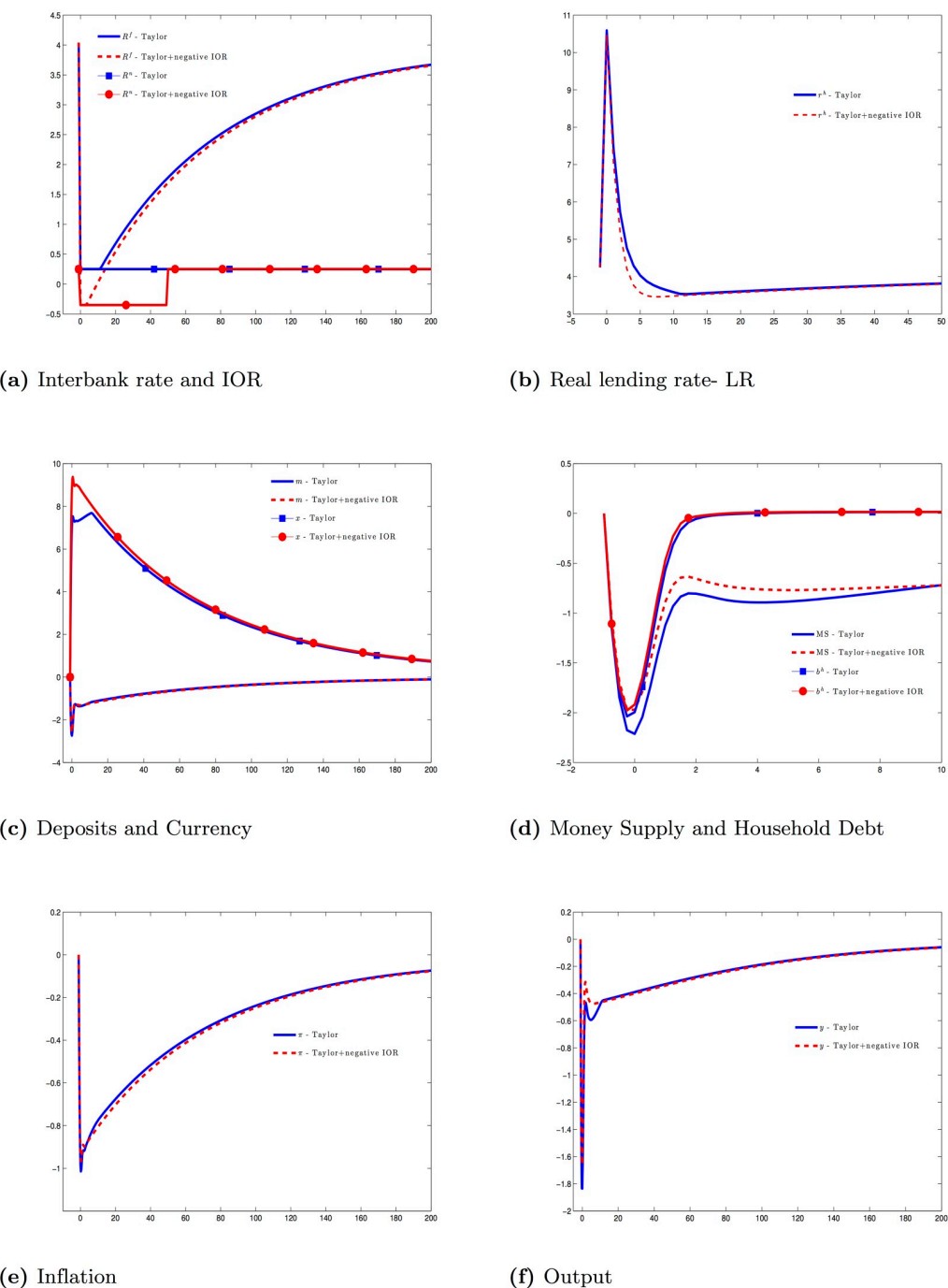

**(a)** Interbank rate and IOR

**(b)** Real lending rate- LR

**(c)** Deposits and Currency

**(d)** Money Supply and Household Debt

**(e)** Inflation

**(f)** Output

**Fig 2. Financial crisis under Taylor rule (M2) and negative IOR (M3).**

characterize a forward guidance policy:

$$R_t^f = \max\left\{ \frac{\overline{\pi}}{\beta} \left( \frac{\pi_{t+1}}{\overline{\pi}} \right)^{\phi_\pi} \exp(\epsilon_t), \quad \overline{R^n} + \delta_f \right\}$$

$$\epsilon_t = \begin{cases} \overline{\epsilon}, & \text{if } t \leq T_{FG} \\ 0, & \text{if } t > T_{FG} \end{cases} \tag{M4}$$

$$R_t^n = \overline{R^n}$$

$$\kappa_t = \rho_\kappa \kappa_{t-1} + (1 - \rho_\kappa)\overline{\kappa}, \quad \kappa_0 \text{ is given}$$

Central bank still follows the above Taylor rule. However, during the forward guidance period $0 \leq t \leq T_{FG}$, central bank commits to lower the intercept of the Taylor rule by $\exp(\overline{\epsilon})$. We set $\overline{\epsilon} = -1/400$. In the previous experiment, the interbank rate is bounded by the IOR during the first 12 periods. As a result of that, we set the horizon for forward guidance as 4 years ($T_{FG} = 16$) to evaluate its efficacy. Fig 3 reflects this monetary policy.

The key channel that a forward guidance affecting a real economy is to increase the expected inflation. Consequently, it lowers the real short-term interbank rate, which in turn passes through to the real lending rate. In comparison to a common Taylor rule, forward guidance is much more effective at raising the aggregate demand.

The effectiveness of a forward guidance policy depends mostly on the inflation path, which is influenced by the path of the interbank rate and the money supply. Fig 3 shows that when the money supply declines sharply, the effectiveness of a forward guidance policy is limited.

## Financial crisis—Mixed rule

The previous section shows that a monetary policy targeting only the interbank rate is not efficient to deal with a banking crisis. What can central bank do to improve the situation? If the main issue is a lack of liquidity in the private sector when banks cut loans, a natural guess should be a policy of targeting the money supply directly. In this section, we examine a modification of a Taylor rule. In normal times, central bank still targets the interbank rate by a Taylor rule. However, in crises, when the interbank rate is near zero and the deflation is still severe, central bank will switch to target the growth of money supply $m_t^s = m_t + x_t$. All the exogenous shocks and monetary policy can be described by the following system of equations:

$$R_t^f = \frac{\overline{\pi}}{\beta} \left( \frac{\pi_{t+1}}{\overline{\pi}} \right)^{\phi_\pi} \left( \frac{m_t^s}{m_{t-1}^s} \right)^{\phi_m}$$

$$R_t^n = \overline{R^n} \tag{M5}$$

$$\kappa_t = \rho_\kappa \kappa_{t-1} + (1 - \rho_\kappa)\overline{\kappa}, \quad \kappa_0 \text{ is given}$$

where $\phi_m = 0.2$ measures the reaction of the interbank rate to the growth of the money supply. The problem (**P5**) is defined to contain the equations in (**AD**) and (M5).

We set $\phi_m$ small relative to $\phi_\pi$ for two reasons. First, it means that in normal times, the interbank rate is not influenced much by the growth rate of the money supply. Second, it implies that, in crises, when lowering the interbank rate is not enough for raising inflation, central bank will respond aggressively by raising the money supply through helicopter money.

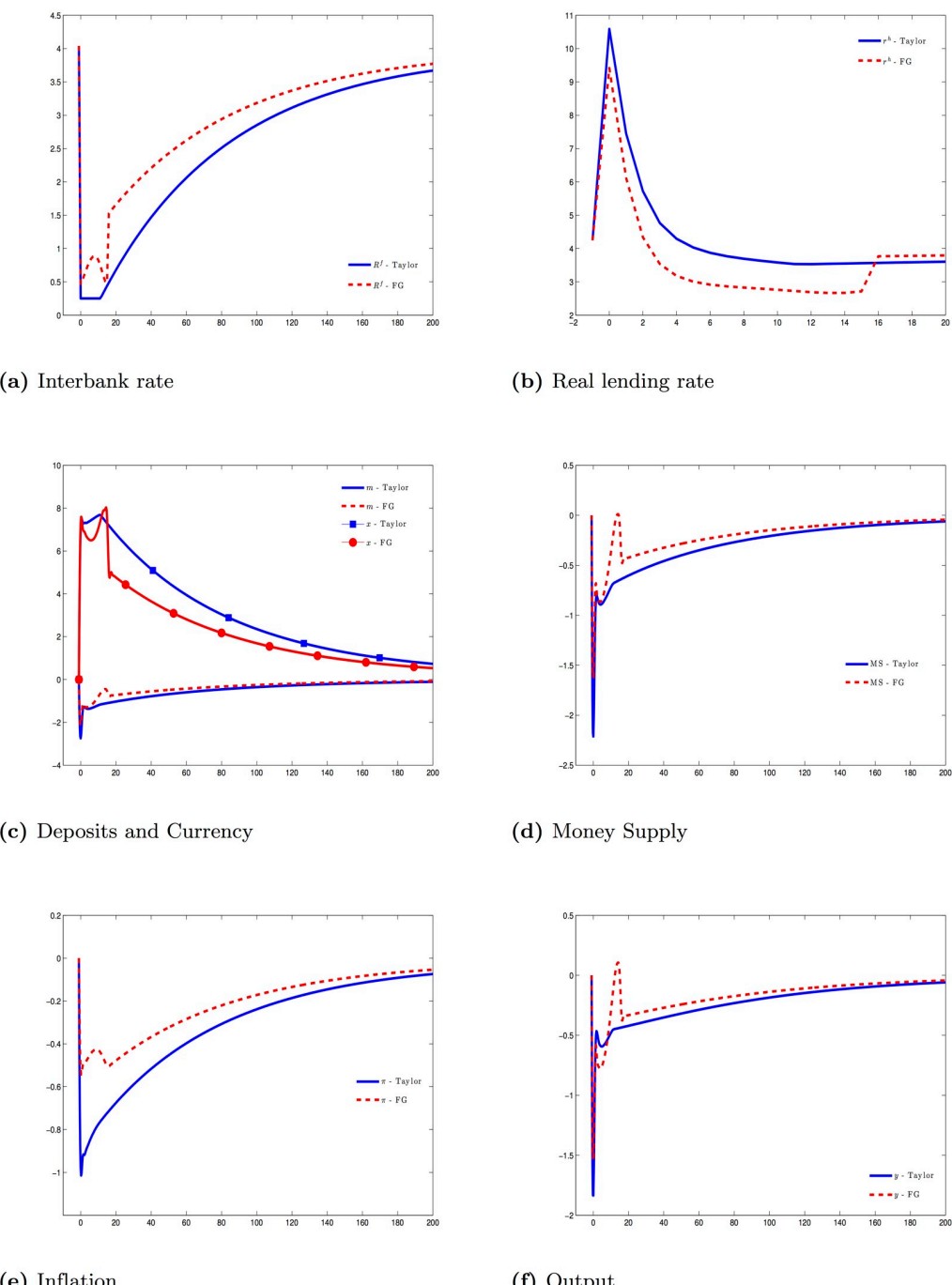

**(a)** Interbank rate

**(b)** Real lending rate

**(c)** Deposits and Currency

**(d)** Money Supply

**(e)** Inflation

**(f)** Output

**Fig 3. Financial crisis—Forward guidance (FG- M4) and Taylor rule (Taylor- M2).**

Why? We can rewrite the modified Taylor rule as:

$$\frac{m_t^s}{m_{t-1}^s} = \left(\frac{R_t^f \beta}{\overline{\pi}}\right)^{1/\phi_m} \left(\frac{\overline{\pi}}{\pi_{t+1}}\right)^{\phi_\pi/\phi_m}$$

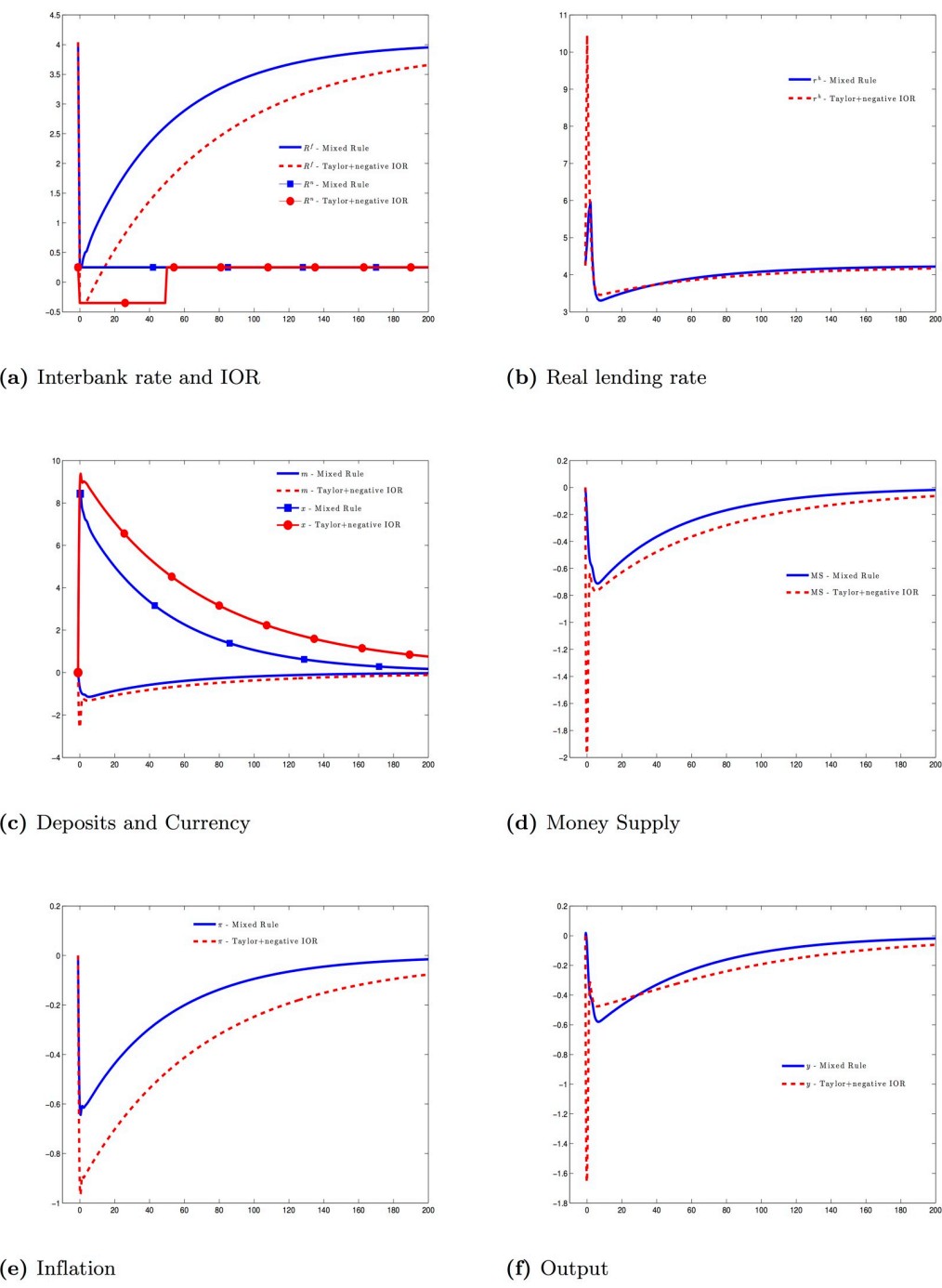

**(a)** Interbank rate and IOR

**(b)** Real lending rate

**(c)** Deposits and Currency

**(d)** Money Supply

**(e)** Inflation

**(f)** Output

**Fig 4. Financial crisis—Mixed rule (mixed rule—M5) and interest rate policy (Taylor+negative IOR—M3).**

When the interbank rate is at its lower bound $R_t^f = R_t^n = \overline{R^n}$, it means that $\phi_\pi/\phi_m$ shows how aggressively central bank will raise the money supply to deal with deflation. Fig 4 shows the economy's response under this modified Taylor rule. Here are some important remarks:

- When targeting the interbank rate is not efficient anymore, switching to target the growth rate of the money supply is very effective. Both inflation and output paths in this experiment

are smoother and less volatile than a Taylor rule with a negative IOR. A modified Taylor rule can anchor inflationary expectations better; thereby restricting the hike in the real lending rate. This result is similar to [33]. Their research also shows that when inflation is out of a bounded region, switching from a Taylor rule to target the growth rate of the money supply can reduce the volatility of the economy.

- The interest rate path alone does not reflect the stance of the monetary policy. If we only look at the paths of the interbank rate, a Taylor rule with a negative IOR keeps the interbank rate not only longer at the lower bound but also lower in every period in comparison to a mixed rule in this section (Fig 4). If we follow the common New Keynesian logic, inflation should have been higher in the previous experiment. However, this is not the case here. When we model explicitly the microfoundation in the banking sector, the link between the money supply and the interest rate is not as tight as the one in the New Keynesian literature with money in the utility function or cash-in-advance. Central bank does not control the interest rate by directly changing the money supply here.

- The inflationary expectation is anchored by both the money supply and the interest rate. Seemly, money supply is a more credible signal for the inflation path in economic crises.

### Taylor rule and Friedman's k-percent rule

The new tool IOR allows central bank to target both the interest rate and the money supply at the same time. In this section, central bank is assumed to follow a Friedman's k-percent rule during our crisis. The following Friedman's k-percent rule indicates that the growth of the money supply is fixed at a constant level:

$$\frac{M_t}{M_{t-1}} = \overline{\pi}$$

At the same time, IOR is adjusted following a Taylor rule. The set of monetary policies and the exogenous shock can be written as:

$$
\begin{cases}
\dfrac{m_t}{m_{t-1}} = \dfrac{\overline{\pi}}{\pi_t}, & \text{if } t \le T_e \\[2ex]
R_t^f = \max\left[\dfrac{\overline{\pi}}{\beta}\left(\dfrac{\pi_{t+1}}{\overline{\pi}}\right)^{\phi_\pi}, \quad \overline{R^n} + \delta_f\right], & \text{if } t > T_e \\[3ex]
R_t^n = \begin{cases}
(1-\alpha_n)\overline{R^n} + \alpha_n\dfrac{\overline{\pi}}{\beta}\left(\dfrac{\pi_{t+1}}{\overline{\pi}}\right)^{\phi_\pi}, & \text{if } t \le T_e \\[2ex]
\overline{R^n}, & \text{if } t > T_e
\end{cases} \\[4ex]
\kappa_t = \rho_\kappa \kappa_{t-1} + (1-\rho_\kappa)\overline{\kappa}, \quad \kappa_0 \text{ is given}
\end{cases}
$$
(M6)

where $\alpha_n = 0.8$ and $T_e = 50$. Together with the equations in (**AD**), (M6) sets up the problem (**P6**). This experiment can be summarized as followings. Before time $T_e$, central bank targets both the IOR and the money supply by, respectively, a Taylor rule and a Friedman's k-percent rule. After time $T_e$, central bank comes back to its Taylor rule alone. Fig 5 compares the effect of monetary polices in (M6) with the previous experiments: Here are some important remarks:

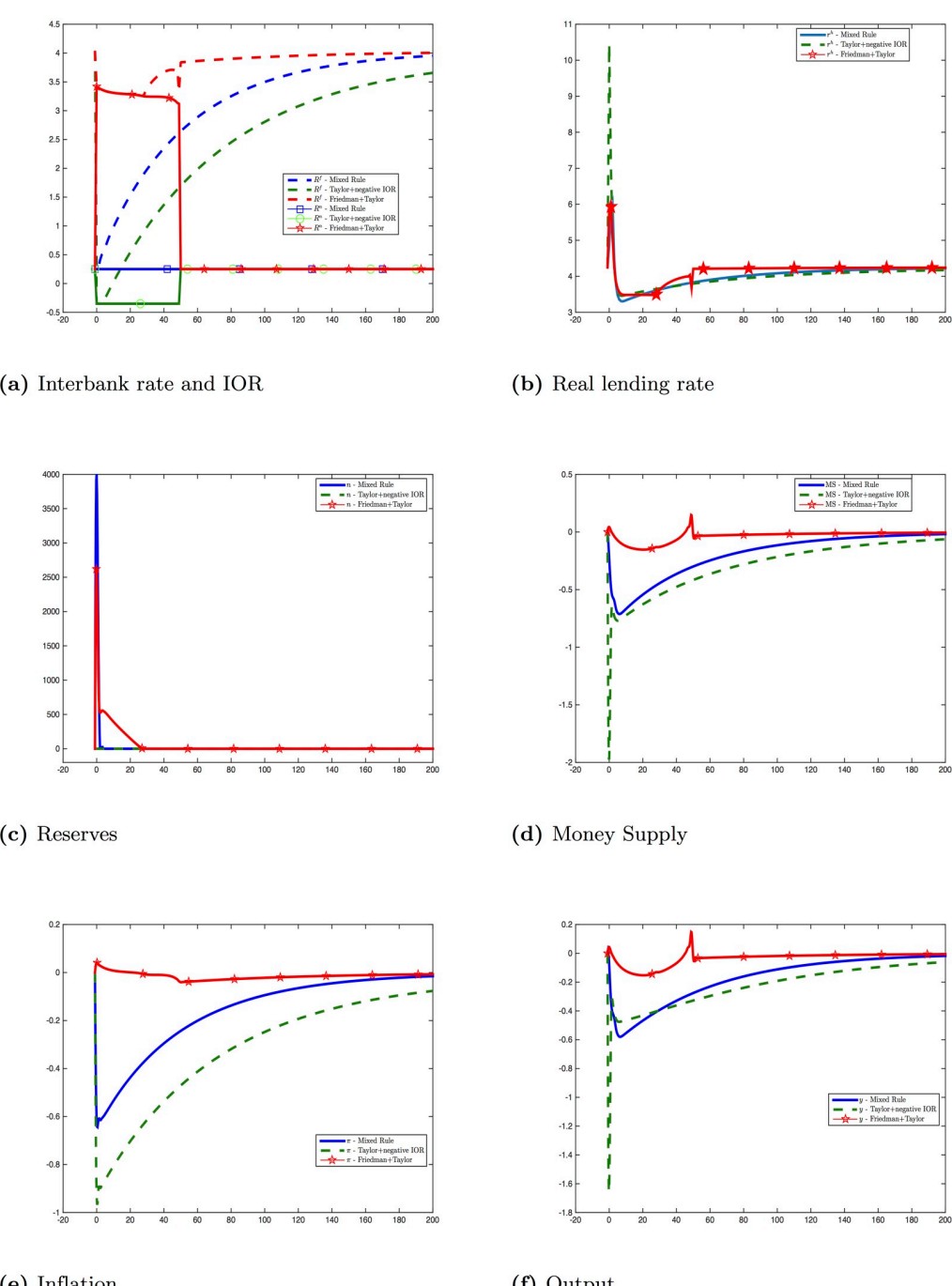

**(a)** Interbank rate and IOR

**(b)** Real lending rate

**(c)** Reserves

**(d)** Money Supply

**(e)** Inflation

**(f)** Output

**Fig 5. Financial crisis—Friedman+Taylor (M6) vs mixed rule (M5) vs Taylor+negative IOR (M3).**

- Targeting both the money supply and the interest rate is extremely efficient. The inflation rate is nearly anchored at the target level for the whole time. As our model does not have any real rigidities, it implies the output is also at the steady state level.

- The byproduct of targeting the growth rate of the money supply is clearly the sharp increase of reserves and excess reserves. Reserves increase by 25 times in our model and the reserve

requirement is no longer binding for 25 periods. Many economists worry that a huge amount of excess reserves might prevent the effectiveness of the monetary policy or create hyperinflation. Our model shows these concerns have no foundations. By using the IOR, central bank can control the interbank rate. The effect is very similar to the one when central bank uses open market operations. There are also no reasons to believe the huge amount of reserves will create a huge amount of money supply. When the reserve requirement is no longer binding, we cannot use the logic in the money multiplier model to create the link between the monetary base and the money supply anymore. Until there is still a borrowing constraint and a capital requirement, the endogenous money supply is always bounded.

- Once again, we emphasize that the stance of monetary policy can only be judged when we observe both the nominal interbank rate, the money supply and the real short-term rate. The money supply and the interest rate can move in any directions. It can be the case that central bank increases the money supply and the IOR at the same time. It might be a serious mistake to infer that it would cause a deflation.

- The above point raises an important issue about central bank's communication. In reality, the stance of the monetary policy, when sending to the public, is often summarized by only one indicator: the interbank rate. Indeed, this is a common and good practice as the interbank rate is a unique short-term target that central bank controls completely. In normal times, it is a good predicator of inflation path. However, the current situation is very tricky. The interbank rate in most developed countries has been near zero for a long time, and the inflation is persistently lower than its target. The growth rate of the money supply should be included as a part of central bank's communication with the public.

## Conclusion

With a huge amount of excess reserves in the banking systems, IOR is now the most crucial tool for central banks in developed countries. This tool opens a new monetary policy framework where central banks can target both interest rate and money supply at the same time. Of course, in normal times, adjusting interbank rate alone is always timely and much more transparent than targeting money supply, which is not entirely controlled by central banks. However, in recent economic crises, our research shows that the link between interbank rate and loan rate is very weak. If central banks simultaneously target both interbank rate and money supply, they can hit the inflation targets better.

### Mathematical appendix

**Proof for Theorem 1**:

We rewrite the Eqs (8)–(10):

$$\gamma_t = \frac{\beta R_t^f \gamma_{t+1}}{\pi_{t+1}} + \mu_t^c \tag{39}$$

$$\gamma_t = \frac{\beta R_t^m \gamma_{t+1}}{\pi_{t+1}} + \mu_t^c + \varphi \mu_t^r \tag{40}$$

$$\gamma_t = \frac{\beta R_t^n \gamma_{t+1}}{\pi_{t+1}} + \mu_t^c + \mu_t^r \tag{41}$$

As $\mu_t^c$ and $\mu_t^r$ are non-negative and $\gamma_t > 0$, we have $R_t^n \leq R_t^m \leq R_t^f$.

The "=" happens when $\mu_t^r = 0$, or when the reserver requirement is no longer binding.

**Proof for Theorem 2**:

Substitute (32), (35) and (36) into the reserve flows Eq (1), we have:

$$\frac{n_{t-1} + x_{t-1}}{\pi_t} + \hat{\tau}_t = n_t + x_t$$

**Proof for Theorem 3**:

We omit the subscript "t" to denote the steady state value of a variable. Under the Assumption (2) and the result of the Theorem (2) $\Rightarrow \pi = \overline{\pi}$. Under the Assumption (2) and the Taylor rule (34) $\Rightarrow R^f = \overline{\pi}/\beta$. Replace this value of $R^f$ into (8) $\Rightarrow \mu^c = 0$ (the capital requirement is not binding). From (10) and (9) $\Rightarrow R^m$; from (11) $\Rightarrow q^l$.

$$\frac{\mu^r}{\gamma} = 1 - \frac{\beta R^n}{\pi}$$

$$R^m = \left(1 - \varphi \frac{\mu^r}{\gamma}\right) \frac{\pi}{\beta}$$

$$q^l = \frac{\beta \delta_b - \theta \pi}{\pi - \beta(1 - \delta_b)}$$

$$\hat{\tau} = \frac{\pi - 1}{\pi}$$

Under the steady state, (31) $\Rightarrow p^m = (\epsilon - 1)/\epsilon$. Next we move on the household's equation and can find the steady state of:

$$(24) \Rightarrow \quad \lambda = \frac{\chi}{p^m}$$

$$(25) \Rightarrow \quad \eta_1 = \left(\lambda - \frac{\tilde{\beta}\lambda}{\pi}\right) \frac{\pi}{\tilde{\beta}}$$

$$(26) \Rightarrow \quad \eta_2 = \left(\lambda - \frac{\tilde{\beta}R^m\lambda}{\pi}\right) \frac{\pi}{\tilde{\beta}R^m}$$

$$(27) \Rightarrow \quad \eta^b = q^l(\lambda + \eta_2) - \frac{\tilde{\beta}[\delta_b + (1 - \delta_b)q^l](\lambda + \eta_2)}{\pi} > 0$$

$$(21) \Rightarrow \quad b^h = \overline{b^h}$$

$$(3) \Rightarrow \quad s = \left(1 - \frac{1 - \delta_b}{\pi}\right) b^h$$

From (23), we can perform $\tilde{c}_i$ as a function of $\tilde{c}$:

$$\tilde{c}_i = \frac{\alpha_i (\tilde{c})^{1-\sigma}}{(\lambda + \eta_i)^{\sigma}} \tag{42}$$

Substituting (42) into the aggregate consumptions (22), we can find the steady state value of $\tilde{c}$, then $\tilde{c}_1$ and $\tilde{c}_2$:

$$\tilde{c} = \left( \sum_{i=1}^{2} \frac{\alpha_i}{(\lambda + \eta_i)^{\sigma-1}} \right)^{\frac{1}{\sigma-1}}$$

All the constraints (15) and (17) are binding:

$$x = \pi \tilde{c}_1$$
$$a = \tilde{c}_2$$

From (5), (32) and (16), we can find $m$ and $n$ from the following equations:

$$\frac{[R^m - (R^n - 1)\varphi]m}{\pi} = a + \frac{\delta_b b^h}{\pi} - \hat{\tau} - q^l s$$
$$n = \varphi m$$

Replacing that into the deposit flows:

$$\sum_{i=1}^{2} c_i = m - \left( \frac{R^m m}{\pi} + q^l s + \theta b^h - \frac{\delta_b b^h}{\pi} - x + \frac{x}{\pi} - \frac{(R^n - 1)n}{\pi} + \tau \right) \tag{43}$$

From (7) and (4) and use (43):

$$c_i = \frac{\alpha_i c^{1-\sigma}}{\gamma^{\sigma}}$$
$$\frac{c^{1-\sigma}}{\gamma^{\sigma}} (\alpha_1 + \alpha_2) = c_1 + c_2$$
$$c_i = \frac{\alpha_i (c_1 + c_2)}{\alpha_1 + \alpha_2}$$
$$c = \left[ \sum_{i=1}^{2} \alpha_i^{1/\sigma} c_i^{\frac{\sigma-1}{\sigma}} \right]^{\frac{\sigma}{\sigma-1}}$$

We know that this steady state only exists if $\kappa$ satisfy the condition such that capital constraint is not binding, so we need the condition:

$$\kappa < 1 - \frac{(1-\varphi)m}{b^h}$$

## System of equations in equilibrium

Bankers:

$$\gamma_t = \left(\frac{\alpha_i c_t}{c_{i,t}}\right)^{1/\sigma} \frac{1}{c_t}, \quad i = 1,2 \tag{44}$$

$$\gamma_t = \frac{\beta R_t^f \gamma_{t+1}}{\pi_{t+1}} + \mu_t^c \tag{45}$$

$$\gamma_t = \frac{\beta R_t^m \gamma_{t+1}}{\pi_{t+1}} + \mu_t^c + \varphi \mu_t^r \tag{46}$$

$$\gamma_t = \frac{\beta R_t^n \gamma_{t+1}}{\pi_{t+1}} + \mu_t^c + \mu_t^r \tag{47}$$

$$(q_t^l + \theta)\gamma_t = \frac{\beta[\delta_b + (1-\delta_b)q_{t+1}^l]\gamma_{t+1}}{\pi_{t+1}} + (1-\kappa)\mu_t^c \tag{48}$$

$$\frac{n_{t-1}}{\pi_t} + \frac{x_{t-1}}{\pi_t} + \hat{\tau}_t = n_t + x_t \tag{49}$$

$$m_t = \frac{R_{t-1}^m m_{t-1}}{\pi_t} + q_t^l s_t + \theta b_t^h - \delta_b \frac{b_{t-1}^h}{\pi_t} + c_{2,t} + c_{1,t} - x_t + \frac{x_{t-1}}{\pi_t} - \frac{(R_{t-1}^n - 1)n_{t-1}}{\pi_t} + \hat{\tau}_t \tag{50}$$

$$0 \le \mu_t^r \perp (n_t - \varphi m_t) \ge 0 \tag{51}$$

$$0 \le \mu_t^c \perp (n_t + (1-\kappa_t)b_t^h - m_t) \ge 0 \tag{52}$$

$$b_t^h = (1-\delta_b)\frac{b_{t-1}^h}{\pi} + s_t \tag{53}$$

Households:

$$\lambda_t + \eta_{i,t} = \left(\frac{\alpha_i \tilde{c}_t}{\tilde{c}_{i,t}}\right)^{1/\sigma} \frac{1}{\tilde{c}_t}, \qquad i = 1, 2 \tag{54}$$

$$p_t^m \lambda_t = \chi \tag{55}$$

$$\lambda_t = \frac{\tilde{\beta}(\lambda_{t+1} + \eta_{1,t+1})}{\pi_{t+1}} \tag{56}$$

$$\lambda_t = \frac{\tilde{\beta} R_t^m (\lambda_{t+1} + \eta_{2,t+1})}{\pi_{t+1}} \tag{57}$$

$$q_t^l(\lambda_t + \eta_{2,t}) = \frac{\tilde{\beta}[\delta_b + (1 - \delta_b)q_{t+1}^l](\lambda_{t+1} + \eta_{2,t+1})}{\pi_{t+1}} + \eta_t^b \tag{58}$$

$$0 \leq \eta_{1,t} \perp \left(\frac{x_{t-1}}{\pi_t} - \tilde{c}_{1,t}\right) \geq 0 \tag{59}$$

$$0 \leq \eta_{2,t} \perp \left(\frac{R_{t-1}^m m_{t-1}}{\pi_t} + q_t^l s_t + \hat{\tau}_t - \frac{(R_{t-1}^n - 1)n_{t-1}}{\pi_t} - \delta_b \frac{b_{t-1}^h}{\pi_t} - \tilde{c}_{2,t}\right) \geq 0 \tag{60}$$

$$0 \leq \eta_t^b \perp (\overline{b^h} - b_t^h) \geq 0 \tag{61}$$

Firms:

$$1 - \iota(\pi_t - \overline{\pi})\pi_t + \iota\tilde{\beta} \frac{\lambda_{t+1}}{\lambda_t}\left(\pi_{t+1} - \overline{\pi}\right)\pi_{t+1} \frac{y_{t+1}}{y_t} = (1 - p_t^m)\epsilon \tag{62}$$

$$y_t = l_t \tag{63}$$

Market Clearing:

$$y_t = \sum_{i=1}^{2}(c_{i,t} + \tilde{c}_{i,t}) + \theta b_t^h + \frac{\iota}{2}(\pi_t - \overline{\pi})^2 y_t \tag{64}$$

$$c_t = \left[\sum_{i=1}^{2} \alpha_i^{\frac{1}{\sigma}} c_{i,t}^{\frac{\sigma-1}{\sigma}}\right]^{\frac{\sigma}{\sigma-1}} \tag{65}$$

$$\tilde{c}_t = \left[\sum_{i=1}^{2} \alpha_i^{\frac{1}{\sigma}} \tilde{c}_{i,t}^{\frac{\sigma-1}{\sigma}}\right]^{\frac{\sigma}{\sigma-1}} \tag{66}$$

Central bank:

$$R_t^f = \max\left\{ \overline{R^f}\left(\frac{\pi_{t+1}}{\overline{\pi}}\right)^{\phi_\pi}, \quad \overline{R^n} + \delta_f \right\} \tag{67}$$

$$R_t^n = \overline{R^n} \tag{68}$$

## Numerical method

**Inequality constraints.** There are 6 occasionally binding inequality constraints in our model: the reserve requirement, the capital requirement, the deposit-in-advance, the cash-in-advance, the household's borrowing constraint and the Taylor rule of central bank. We deal with these occasionally binding constraints by three methods:

**Method 1**: We apply the method in [34, 35] to transform the complementary conditions into the equality constraints. For example, if we have the following complementary condition:

$$\mu_t f(x_t) = 0, \quad \mu_t \geq 0, \quad f(x_t) \geq 0$$

We create a new variable $\underline{\mu}_t$ and transforms the above conditions into two following equations:

$$\mu_t = \max\{\underline{\mu}_t, 0\}^3$$
$$f(x_t) = \max\{-\underline{\mu}_t, 0\}^3$$

We apply this method for the reserve requirement, deposit-in-advance and cash-in-advance constraints.

**Method 2**: For the capital requirement and the household's borrowing constraint, we apply the penalty method in [36] to avoid the ill-conditioned of the system and deal with occasionally binding constraints. So the utility of banker and the capital constraint will be changed as:

$$U = \log c_t - \frac{\rho_e}{4}\max\{\underline{\mu}_t^c, 0\}^4$$
$$n_t + b_t^f + (1 - \kappa_t)b_t^h - m_t = -\underline{\mu}_t^c$$

where $\rho_e = 1e6$ is the penalty coefficient. When the capital constraint is violated, banker will lose the utility. However, when they get positive net worth, they do not get reward for that. The household's utility also is changed to deal with the borrowing constraint.

**Method 3**: For the Taylor rule of central bank, we use the soft max constraint to deal with the lower bound on $R_{min}^f = \overline{R^n} + \delta_f$ so we can still take derivative to solve the system of equations:

$$u_t = \overline{R^f}\left(\frac{\pi_{t+1}}{\overline{\pi}}\right)^{\phi_\pi}$$

$$R_t^f = \begin{cases} u_t + \frac{\log(1+\exp(s_{max}(R_{min}^f - u_t)))}{s_{max}}, & \text{if } u_t \geq R_{min}^f \\ R_{min}^f + \frac{\log(1+\exp(s_{max}(u_t - R_{min}^f)))}{s_{max}}, & \text{if } u_t < R_{min}^f \end{cases}$$

When $s_{max} \to \infty$, the soft max constraint converges to the hard max constraint. We choose the coefficient $s_{max} = 1e4$.

**Dynamics of economy.** We solve the perfect foresight equilibrium with the unexpected shock by assuming that after $T = 300$ quarters, the economy will converge back to the initial steady state. The initial position before the unexpected shocks is the steady state. Basically, we need to solve a large system of equations to determine the dynamic path of the economy. The transform of occasionally inequality constraints in the previous section ensures that every equation is continuous and differentiable.

For every application, we use homotopy method for solving this large system of equation, with the initial point starting from the steady state or the previous result. We use Ipopt written by [37] with the linear solver HSL (a collection of Fortran codes for large scale scientific computation. http://www.hsl.rl.ac.uk/) to conduct homotopy.

In all cases, we can solve successfully the perfect foresight path with the accuracy at least $1e - 9$. In some results, only at the last period $T - 1$, we observe that there is a big switch from deposits to currency while the total money supply does not change much. When we increase the length of periods from $T = 300$ to $T = 1000$ the path does not change. We omit the result in the last period to ensure the smoothness of the result.

## Author Contributions

**Conceptualization:** Duong Ngotran.

**Methodology:** Duong Ngotran.

**Writing – original draft:** Duong Ngotran.

**Writing – review & editing:** Duong Ngotran.

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
