## [Decision Letter · Decision Letter 0]

28 Jan 2021

PONE-D-20-26141

Interest on reserves, helicopter money and new monetary policy

PLOS ONE

Dear Dr. Ngotran,

Thank you for submitting your manuscript to PLOS ONE. After careful consideration, we feel that it has merit but does not fully meet PLOS ONE’s publication criteria as it currently stands. Therefore, we invite you to submit a revised version of the manuscript that addresses the points raised during the review process.

After you have revised your manuscript, please, let a professional proofread it. Personally, I recommend that you use American Journal Experts (aje.com).

We look forward to receiving your revised manuscript.

Kind regards,

Mikael Bask

Academic Editor

PLOS ONE

Journal Requirements:

3. We note you have included a table to which you do not refer in the text of your manuscript. Please ensure that you refer to Tables 3 and 7 in your text; if accepted, production will need this reference to link the reader to the Table.

4. Thank you for submitting the above manuscript to PLOS ONE. During our internal evaluation of the manuscript, we found significant text overlap between your submission and the following works, of which you are an author:

- https://mpra.ub.uni-muenchen.de/81579/1/MPRA_paper_81579.pdf

- https://www.econstor.eu/bitstream/10419/203245/1/1676918205.pdf

Could you please indicate if either of these works have been peer reviewed or formally published?

Reviewers' comments:

Reviewer's Responses to Questions

**Comments to the Author**

1. Is the manuscript technically sound, and do the data support the conclusions?

Reviewer #1: No

Reviewer #2: Yes

2. Has the statistical analysis been performed appropriately and rigorously? 

Reviewer #1: No

Reviewer #2: N/A

3. Have the authors made all data underlying the findings in their manuscript fully available?

Reviewer #1: No

Reviewer #2: Yes

4. Is the manuscript presented in an intelligible fashion and written in standard English?

Reviewer #1: No

Reviewer #2: Yes

5. Review Comments to the Author

Reviewer #1: The paper addresses an important economic question even more so during the current crisis, where countries may need to increase the money supply to finance unusually large fiscal expansions. However, I think the paper has a lot to be desired to be considered further for publication.

The key question is what are the new lessons from considering an electronic money economy. In this respect, while the author performs a series of exercises, it is hard to see what is the main punchline and what is the main novelty relatively to previous studies. In general, I believe the paper should be streamlined or re-focused on the key-issue, which is to illustrate the implications of electronic money for monetary policy.

1. The literature review has developed in a non-technical way. What does it make the paper similar to the cited literature? And, what does it make the paper different from the cited literature? Electronic money is far from a dense literature in DSGE models. In order to introduce that analysis in the literature on monetary policy, it is needed to be more precise. It is not obvious why monetary policy should take account of electronic money. The author needs to work on his introduction and his related literature sections much more seriously to address these concerns.

2. The author describes the model in too much detail without a clear message. The author puts forward the following three results:

a. Following a Taylor, it is not efficient;

b. Negative interest on reserves or forward guidance is effective;

c. Targeting both interest rate and money supply by a Taylor rule and a Friedman’s k-percent rule, inflation and output are stabilized.

I think that these results can be interesting when thought about separately, but I suggest that the author takes a stand on one main message that he would want to deliver through this paper. Is the paper suggesting that the money supply target is the most desirable out of the policies analyzed? Alternatively, is this paper analyzing the dependence of Taylor rule effectiveness on the interaction with other policies (e.g. money supply targeting)? On the other hand, is this paper about comparing the effectiveness of different monetary policies in presence of electronic money? Once I am convinced with the motivation of this paper, I think choosing any of these three questions would lead to nice contributions. However, crucially, the paper should stick to one question and really nail an answer, before moving onto the next question.

3. The model section should be rewritten by adhering to community standards for DSGE model.

a. Household;

b. Firms (retail and wholesale branches);

c. Banks;

d. Government.

The author should precise what are the components of the model borrowed by the literature and what are those added by the author itself – and, for which reason he adds them.

4. He should clarify why it is crucial taking account of electronic money in order to compare the effectiveness of monetary policies.

Reviewer #2: Reviewer Report

Title: “Interest on reserves, helicopter money and new monetary policy”

Summary: This paper studies the effects of several monetary policy tools (interest on reserves and monetary base) using a nonlinear dynamic model with currency, demand deposits and bank reserves.

I found this paper to be on an important topic and has interesting results. Below I make some comments for improvement.

Main comments:

The paper is fairly well written in the sense that it is clear about what it does. But there are many grammar mistakes. I would suggest having it proof-read.

I found section 2 to be very confusing. I actually think the paper is easier to understand if section 2 is removed and the reader goes straight to section 3. This has the advantage of avoiding repeating equations as in the current version. Any important content in section 2 can be incorporated in section 3.

Remove Tables 1 to 6. I found them confusing and not very informative.

I found the calibration of loan amortization (δb) odd. In section 2.4 it is said that if an agent borrows £1 then the repayment will be:

δ_b in year 1

δ_b x (1-δ_b) in year 2

δ_b x〖(1-δ_b)〗^2 in year 3

and so on…

If one sums the numbers from year 2 onwards then we get:

Sum=δ_b×(δ_b)/((1-δ_b) )

With δ_b set at 0.5 then we get the above is 0.5.

Even if we were to add a discount factor to the above (say β) it would not change much.

Compare this to a perpetuity which has present value of 1/r. If we set r=0.04 then we get 25.

The loan amortization seems to me to be implying an excessively high interest rate (or maybe there is something I am missing here…). I would therefore suggest adopting a different value for δ_b.

6. PLOS authors have the option to publish the peer review history of their article (what does this mean?). If published, this will include your full peer review and any attached files.

Reviewer #1: No

Reviewer #2: No

---

## [Author Response · Author response to Decision Letter 0]

21 Mar 2021

Response to Reviewer 1 on “Interest on reserves, helicopter money and new monetary policy” for PLOS ONE

I thank the referee for reading my work and providing thoughtful and fair comments on it. Below I will outline how I would respond to the referees’ suggestions:

General comments: “The key question is what are the new lessons from considering an electronic money economy. In this respect, while the author performs a series of exercises, it is hard to see what is the main punchline and what is the main novelty relatively to previous studies. In general, I believe the paper should be streamlined or re-focused on the key-issue, which is to illustrate the implications of electronic money for monetary policy”.

My Response:

I am happy to do this in my revised version by emphasizing the contribution of my paper (Introduction- page 2 line 29). With electronic money and interest on reserves, central banks could “simultaneously” move money supply and interest rate in every direction. In textbooks, when money supply increases, interest rate goes down. However, in the electronic monetary system, if central banks want, interest rate could go up when money supply increases. 

1. “The literature review has developed in a non-technical way. What does it make the paper similar to the cited literature? And, what does it make the paper different from the cited literature? Electronic money is far from a dense literature in DSGE models. In order to introduce that analysis in the literature on monetary policy, it is needed to be more precise. It is not obvious why monetary policy should take account of electronic money. The author needs to work on his introduction and his related literature sections much more seriously to address these concerns”.

My Response: Like my answer in the previous comment, I will definitely emphasize why monetary policy should take account of electronic money in the introduction (Introduction-page 2 line 29 ). 

For the literature review, as there was no dense literature on monetary policy with electronic money, I tried linking my model with most relevant studies. In my revised version (Introduction-page 2 line 62), I will start from the closet literature- New Keynesian. Then how my model incorporates other elements to model an economy with electronic money properly.

2. “The author describes the model in too much detail without a clear message. The author puts forward the following three results:

a. Following a Taylor, it is not efficient;

b. Negative interest on reserves or forward guidance is effective;

c. Targeting both interest rate and money supply by a Taylor rule and a Friedman’s k-percent rule, inflation and output are stabilized.

I think that these results can be interesting when thought about separately, but I suggest that the author takes a stand on one main message that he would want to deliver through this paper. Is the paper suggesting that the money supply target is the most desirable out of the policies analyzed? Alternatively, is this paper analyzing the dependence of Taylor rule effectiveness on the interaction with other policies (e.g. money supply targeting)? On the other hand, is this paper about comparing the effectiveness of different monetary policies in presence of electronic money? Once I am convinced with the motivation of this paper, I think choosing any of these three questions would lead to nice contributions. However, crucially, the paper should stick to one question and really nail an answer, before moving onto the next question”.

My Response: I am happy to explain this important point in my abstract and introduction. Traditional monetary policy only cares on either interest rate or money supply policy separately. Currently, the dominant New Keynesian framework only characterizes a monetary policy by an interest rate rule. My paper shows that an interest rate rule could be in company with a money supply targeting.

 I will insert the following sentence in my abstract:

“An interest rate rule policy alone is just a subset of a more general monetary policy framework when central bank can move interest rate and money supply in every direction.”

In my conclusion, I will emphasize the support of my paper on the combination of a Taylor rule and a Friedman k-percent rule.

3. “The model section should be rewritten by adhering to community standards for DSGE model.

a. Household;

b. Firms (retail and wholesale branches);

c. Banks;

d. Government.

The author should precise what are the components of the model borrowed by the literature and what are those added by the author itself – and, for which reason he adds them”.

My Response: I insert this following sentence at the literature review (page 3- line 83)

“The model is very similar to a standard New Keynesian model, except the flows of electronic payments and the banker problem”. 

I have to add bankers and reserves into this model to understand the connection between interest on reserves, interbank rate and money supply. 

4. “He should clarify why it is crucial taking account of electronic money in order to compare the effectiveness of monetary policies”.

My Response: I am happy to do this in my revised version by emphasizing the contribution of my paper (Introduction-page 2 line 29).

Response to Reviewer 2 on “Interest on reserves, helicopter money and new monetary policy” for PLOS ONE

I thank the referee for a positive feedback and providing thoughtful and fair comments on it. Below I will outline how I would respond to the referees’ suggestions:

1. “I found section 2 to be very confusing. I actually think the paper is easier to understand if section 2 is removed and the reader goes straight to section 3. This has the advantage of avoiding repeating equations as in the current version. Any important content in section 2 can be incorporated in section 3.

Remove Tables 1 to 6. I found them confusing and not very informative”.

My Response:

I separate section 2 from section 3 to separate the environment of the model and each agent’s optimal problem. If my paper followed a standard New Keynesian model, Section 2 would not be necessary. However, transactions with reserves and deposits are not dense in the literature. I hope that my paper will be easily understood for all non-experts, so I trade off between the paper length and details of the environment.

I insert the Tables 1 to 6 for the same reason. Tables 1 to 6 are unnecessary for economists working at central banks or economists specializing in monetary policy. However, many economists who do not fall into the above categories could find them useful. In the case that the editor agrees with the referee, I am happy to move them to the Appendix.

2. “I found the calibration of loan amortization (δb) odd. In section 2.4 it is said that if an agent borrows £1 then the repayment will be:

δ_b in year 1

δ_b x (1-δ_b) in year 2

δ_b x〖(1-δ_b)〗^2 in year 3

and so on…

If one sums the numbers from year 2 onwards then we get:

Sum=δ_b×(δ_b)/((1-δ_b) )

With δ_b set at 0.5 then we get the above is 0.5.

Even if we were to add a discount factor to the above (say β) it would not change much.

Compare this to a perpetuity which has present value of 1/r. If we set r=0.04 then we get 25.

The loan amortization seems to me to be implying an excessively high interest rate (or maybe there is something I am missing here…). I would therefore suggest adopting a different value for δ_b”.

My Response: The nominal sum of loan payment will be equal to 1 if the nominal amount of borrowing is 1:

δ^b+δ^b (1-δ^b )+δ^b (1-δ^b )^2+⋯.+δ^b (1-δ^b )^n+⋯.

=δ^b [1+(1-δ^b )+(1-δ^b )^2+⋯.+(1-δ^b )^n+⋯]

= δ_b/(1-(1-δ^b ) )

=1

Furthermore, the price of loan ql (<1 in the steady state). When banks give loans to households, the nominal amount of loan is $1, but households only receive $ ql. If we calculate the interest of the loan, it will be equal to rh (equation 12), which is equal around to 4.25% in the steady state.

3. “The paper is fairly well written in the sense that it is clear about what it does. But there are many grammar mistakes. I would suggest having it proof-read”.

My Response: I tried fixing grammar mistakes as much as possible in my paper. I also asked one of my colleagues in English department to help me. Hopefully, this revised version could reduce the errors to the minimum level.

---

## [Editor Report · Decision Letter 1]

22 Apr 2021

PONE-D-20-26141R1

Interest on reserves, helicopter money and new monetary policy

PLOS ONE

Dear Dr. Ngotran,

Thank you for submitting your manuscript to PLOS ONE. After careful consideration, we feel that it has merit but does not fully meet PLOS ONE’s publication criteria as it currently stands. Therefore, we invite you to submit a revised version of the manuscript that addresses the points raised during the review process.

The paper must be proofread by professionals. I strongly recommend the use of AJE (aje.com).

We look forward to receiving your revised manuscript.

Kind regards,

Mikael Bask

Academic Editor

PLOS ONE

Journal Requirements:

Additional Editor Comments (if provided):

The paper needs to be proofread before publication. I recommend the author to use AJE (aje.com).

---

## [Author Response · Author response to Decision Letter 1]

30 May 2021

I thank the editor and two referees for feedbacks on my manuscript. I incorporated these feedbacks into my revised version.

---

## [Editor Report · Decision Letter 2]

17 Jun 2021

Interest on reserves, helicopter money and new monetary policy

PONE-D-20-26141R2

Dear Dr. Ngotran,

We’re pleased to inform you that your manuscript has been judged scientifically suitable for publication and will be formally accepted for publication once it meets all outstanding technical requirements.

Kind regards,

Mikael Bask

Academic Editor

PLOS ONE
---

## [Editor Report · Acceptance letter]

7 Jul 2021

PONE-D-20-26141R2 

Interest on reserves, helicopter money and new monetary policy 

Dear Dr. Ngotran:

I'm pleased to inform you that your manuscript has been deemed suitable for publication in PLOS ONE. Congratulations! Your manuscript is now with our production department. 

Kind regards, 

on behalf of

Dr. Mikael Bask 

Academic Editor

PLOS ONE